# Evaluation of SARS-CoV-2-Specific IgY Antibodies: Production, Reactivity, and Neutralizing Capability against Virus Variants

**DOI:** 10.3390/ijms25147976

**Published:** 2024-07-21

**Authors:** Jacob Schön, Andrea Aebischer, Nico Joël Halwe, Lorenz Ulrich, Donata Hoffmann, Sven Reiche, Martin Beer, Christian Grund

**Affiliations:** 1Institute of Diagnostic Virology, Friedrich-Loeffler-Institut, Federal Research Institute for Animal Health, 17493 Greifswald, Germany; jacob.schoen@fli.de (J.S.); nico.halwe@fli.de (N.J.H.); lorenz.ulrich@fli.de (L.U.); donata.hoffmann@fli.de (D.H.); martin.beer@fli.de (M.B.); 2Department of Experimental Animal Facilities and Biorisk Management, Friedrich-Loeffler-Institut, Federal Research Institute for Animal Health, 17493 Greifswald, Germany; andrea.aebischer@fli.de (A.A.); sven.reiche@fli.de (S.R.)

**Keywords:** antibodies, SARS-CoV-2, IgY, yolk, lumazine synthase, SpyTag/SpyCatcher

## Abstract

The emergence of SARS-CoV-2 in late 2019 initiated a global pandemic, which led to a need for effective therapeutics and diagnostic tools, including virus-specific antibodies. Here, we investigate different antigen preparations to produce SARS-CoV-2-specific and virus-neutralizing antibodies in chickens (n = 3/antigen) and rabbits (n = 2/antigen), exploring, in particular, egg yolk for large-scale production of immunoglobulin Y (IgY). Reactivity profiles of IgY preparations from chicken sera and yolk and rabbit sera were tested in parallel. We compared three types of antigens based on ancestral SARS-CoV-2: an inactivated whole-virus preparation, an S1 spike-protein subunit (S1 antigen) and a receptor-binding domain (RBD antigen, amino acids 319–519) coated on lumazine synthase (LS) particles using SpyCather/SpyTag technology. The RBD antigen proved to be the most efficient immunogen, and the resulting chicken IgY antibodies derived from serum or yolk, displayed strong reactivity with ELISA and indirect immunofluorescence and broad neutralizing activity against SARS-CoV-2 variants, including Omicron BA.1 and BA.5. Preliminary in vivo studies using RBD–lumazine synthase yolk preparations in a hamster model showed that local application was well tolerated and not harmful. However, despite the in vitro neutralizing capacity, this antibody preparation did not show protective effect. Further studies on galenic properties seem to be necessary. The RBD–lumazine antigen proved to be suitable for producing SARS-CoV-2 specific antibodies that can be applied to such therapeutic approaches and as reference reagents for SARS-CoV-2 diagnostics, including virus neutralization assays.

## 1. Introduction

The emergence of severe acute respiratory syndrome coronavirus type 2 (SARS-CoV-2) in late 2019 led to a global pandemic [1], creating a need for effective therapeutics and diagnostic tools [2,3,4]. Despite the availability of vaccines, specific and affordable treatments remain crucial. SARS-CoV-2 specific and virus-neutralizing antibodies have potential as non-invasive therapeutics, such as nasal sprays, and are essential for diagnostic assays like enzyme-linked immunosorbent assay (ELISA) and indirect immunofluorescence assay (IFA).

Chicken egg yolk antibodies, known as immunoglobulin Y (IgY), offer a valuable alternative to mammalian antibodies [5]. Chickens naturally produce IgY in their serum and transfer substantial amounts to egg yolks, providing passive immunity to offspring [6]. Harvested IgY antibodies from yolk can sum up to around 30–45 g annually per hen, of which 2–10% are antigen-specific. Moreover, utilizing chicken eggs for antibody production is less invasive and can yield large quantities of specific antibodies over extended periods [5,7,8], while being very stable [9,10,11]. Unlike mammalian immunoglobulin G (IgG), IgY does not activate the complement system, avoiding inflammatory responses and assay interference [12]. Nevertheless while IgY antibodies offer several advantages such as reduced risk of antibody-dependent enhancement (ADE) and lack of interaction with mammalian Fc receptors [12], these can also limit their effectiveness by reducing immune system engagement [12], leading to a potentially shorter half-life, and impairing their ability to clear pathogens effectively [12]. In addition, IgY is very stable and has good storing properties [9,10], while it also can be freeze-dried [11].

In this study, we evaluated different SARS-CoV-2 antigens to induce specific antibodies in chickens and rabbits. We focused on a recombinant protein representing the receptor-binding domain (RBD, amino acids 319–519) coated on lumazine synthase (LS) particles using the SpyCatcher/SpyTag technology [13,14]. Beside the reactivity of induced antibodies in a variety of diagnostic tests, a special focus of the evaluation was the capacity to neutralize SARS-CoV-2 in vitro and in vivo. RBD–lumazine synthase-induced antibodies in yolk preparations proved to be the most valuable reagent for SARS-CoV-2 diagnostic assays including the in vitro neutralization test. While further optimization is needed for possible local therapeutic applications, IgY from yolk preparations remains a promising non-invasive therapeutic option due to their stability, large-scale production capabilities, and broad-spectrum neutralization potential.

## 2. Results

### 2.1. Chickens Develop High and Stable SARS-CoV-2-Specific Antibody Levels in Sera and Yolk

To generate a SARS-CoV-2 full-virus reference antigen, VeroE6 cells were inoculated with wild type (WT) SARS-CoV-2 D614G (B.1) isolate Germany/BavPat1/2020 (EPI_ISL_406862). After three days of incubation the virus-containing supernatant was harvested and inactivated. In comparison, the RBD and S1 domains of the spike glycoprotein (aa 319–519 and aa 17–685, respectively) were selected as suitable subunit antigens. To optimizes their immunogenicity, the recombinant proteins were tagged with a SpyTag and presented using a modular vaccine platform based on LS [15]. The proteins were heterogeneously expressed in mammalian Expi293 cells, purified from the cell culture supernatant and subsequently conjugated to the pre-assembled LS scaffold using the SpyCatcher/SpyTag technology, forming multimeric protein scaffold paIgrticle (MPSP). Chickens were intramuscularly immunized with the inactivated virus and the RBD- and S1-LS-MPSPs (Figure 1A). These MPSPs subunit antigens were additionally used for the immunization of rabbits, to evaluate a potential species-specificity of the immune response (Figure 1A). The IgY containing chicken yolk was harvested (Figure 1A) and was examined by SARS-CoV-2 RBD-ELISA [16], by IFA and by the virus neutralization test (VNT) and finally for its protective potential in an in vivo hamster experiment.

All antigens utilized for immunization (whole-virus, RBD-MPSP and S1-MPSP) elicited antibody production in sera following the initial immunization, as evidenced by RBD-specific ELISA (Figure 1B). The antibody levels induced by the recombinant antigens after the first immunization were found to be already close to the plateau reached after one subsequent booster immunization on day 21, over a total period of 196 days. The mean plateau OD450nm for RBD was 2.48, while that for S1 was 2.26. In contrast, the booster immunization on day 28 with whole-virus antigen resulted in a pronounced increase in the antibody response. Subsequent booster immunizations on days 49, 71, and 112 with whole-virus antigen had only a slight impact on antibody reactivity, as tested by RBD-ELISA. It is noteworthy that the pronounced increase in antibody levels, as observed by nucleocapsid (N)-specific ELISA, became evident only after the second booster immunization on day 49 (Figure 1B). The reactivity of rabbit sera, four weeks following the second boost, was substantially lower than in chickens, with little difference between RBD and S1 (Figure 1B).

Subsequent to the final immunization of the chickens (day 49 for recombinant proteins and day 112 for the whole-virus group), eggs from the various groups were collected, and the yolks were pooled on a weekly basis, as illustrated in Figure 1A. All initial yolk preparations exhibit robust antibody reactivity, as indicated in Figure 1C. Interestingly, although S1- and RBD-antigen-specific sera exhibited comparable reactivity in the RBD-ELISA (mean OD450nm of 2.43 for S1 and 2.33 for RBD), the reactivity of the yolk samples differed. The reactivity of the S1 yolk preparation exhibited a reactivity level comparable to that of the sera (mean OD450nm 2.02), whereas the reactivity of the RBD-yolk samples was higher than that of the sera samples (mean OD450nm 3.03) (details in S1 table). A comparison of the ELISA results revealed that the reactivity of the yolk samples from chickens immunized with whole virus was distinctly lower in the N-ELISA than in the RBD-ELISA.

The data presented herein demonstrate that all three SARS-CoV-2 antigens were immunogenic in chickens, with a distinct response to the recombinant S1- and RBD-coated MPSP already evident after the first immunization. Booster immunizations induced stable antibody levels of over 100 days, with efficient transport into the yolk, which was superior for the RBD group.

### 2.2. Sera and Yolk Antibodies of RBD-Immunized Chickens Are Reacting Specifically in Indirect Immunofluorescence Test

To assess the ability of yolk antibody preparations to recognize the SARS-CoV-2 antigen within virus-infected cells, IFA were conducted. Yolk pools from the second week of collection and the corresponding sera samples from individual animals were tested to label the wild type SARS-CoV-2 isolate on VeroE6 cells.

All sera demonstrated reactivity with SARS-CoV-2 epitopes presented in the context of the whole virus, resulting in specific signals within infected VeroE6 cells in the immunofluorescence assay (Figure 2A). Antibodies induced by the RBD were localized in the cytoplasm of infected cells, with no presence in the nuclei. The RBD yolk samples exhibited a similar specific signal. Sera specific to S1 also displayed the same distribution pattern, albeit at a much lower level. The signal with yolk-derived antibodies was unclear. Sera raised against the whole virus showed a broader but less-specific signal.

In accordance with the respective sera, yolk samples from the RBD and whole-virus groups successfully stained for virus antigen, although the whole-virus group had more non-specific background staining (Figure 2B). The yolk from the S1-immunized group showed no reactivity. In addition, yolk from the RBD-immunized group was tested against Delta- and Omicron BA.1-infected VeroE6 cells, showing specific reactivity to these variants, as well (Figure 2C).

In summary, differences were observed between the antigens in the immunofluorescence assay. The RBD samples showed the strongest and most-specific signal, even with antigenically distant variants such as Delta and Omicron BA.1, while avoiding reactivity with host cell proteins present in the whole-virus preparation.

To evaluate the potential of the yolk antibody preparations to recognize SARS-CoV-2 antigen in the context of virus-infected cells, the antibodies were tested by immunofluorescence assay. Therefore, yolk pools from the second week of collection and corresponding sera samples from individual animals were tested for staining of WT SARS-CoV-2 isolate on VeroE6 cells.

### 2.3. Sera and Yolk Antibodies of RBD Antigen-Immunized Chickens Are Neutralizing SARS-CoV-2, Including a Broad Spectrum of VOCs

To further evaluate the reactivity and biological properties of the generated antibodies, we performed live-virus neutralization experiments and evaluated the cross-reactivity profile of the different antibody preparations against different SARS-CoV-2 variants.

Antibodies induced by the RBD antigen, as well as by the whole-virus antigen, were able to efficiently neutralize the homologous WT SARS-CoV-2 strain efficiently and this reactivity was maintained for several weeks (Figure 3A). However, neutralizing capacity was only detectable after booster immunization, i.e., at 49 days post primary immunization (dpi). In contrast, VNT titers in chickens immunized with recombinant S1 antigen were weak, only sporadically detectable for one chicken with a titer of 128 at 63 dpi (Figure 3A). This is surprising, because all S1-immunized chickens seroconverted after the first immunization when tested by ELISA (Figure 1B). Animals immunized with the whole-virus preparation showed similar kinetics to those immunized with the RBD, but with lower titers, for the last two time points (Figure 3A). 

This phenomenon is reflected in the testing of yolk samples: only preparations from chickens immunized with RBD antigen and whole-virus antigen had stable VNT titers, whereas yolk preparations from S1-immunized chickens were VNT negative (Figure 3B).

Furthermore, different antibody preparations were subjected to virus neutralization tests against various SARS-CoV-2 variants (Figure 3C). The RBD-immunized group exhibited high neutralization titers against the homologous WT strain (831.7 to 2521.4) and a similar level against the Alpha, Beta and Delta variant (512 to 5042.8), with titer differences with respect to the homologous WT strain below two log2 steps for the Alpha, Beta and Delta variant (ΔVNT (log2), Table 1). Two sera demonstrated reactivity against Omicron BA.1, with titers of 256 and 831.7, while the serum from the third animal exhibited a loss of detectable humoral neutralizing capacity. Against Omicron BA.5, two animals lost neutralizing activity, while the neutralizing titer of sera of the third animal maintained a high level (415.9), with ΔVNT (log2) of 2.7. Similarly, the pooled yolk of these animals demonstrated the capacity to neutralize all tested virus variants, albeit with slightly diminished efficacy. However, it retained reactivity with all variants (Figure 3D, Table 1).

Similarly, sera from rabbits immunized with the RBD demonstrated the capacity to neutralize all variants (VNT = 512 to 4096), with lowest titers against Omicron BA.5.

Additionally, sera and yolk preparations from chickens immunized with a whole-virus preparation demonstrated an overall broad humoral neutralizing reactivity against all tested variants, albeit at a lower level than the RBD group (Figure 3C,D, Table 1). In contrast sera from S1-immunized animals had only limited neutralizing capacity, with only one of the three chickens exhibiting reactivity with WT (7), Alpha (8.7), and Delta (5.7) variants, but not against the Beta or Omicron BA.1/BA.5 variants (Figure 3C, Table 1). The reactivity was further diminished, exhibiting only minimal neutralizing capacity against the Alpha variant (4.7) (Figure 3D, Table 1). Of the two immunized rabbits, only one exhibited a neutralizing capacity within a range of 5–5.7, but was tested negative against both the Delta and Omicron BA.5 (Figure 3C, Table 1).

Furthermore, the capacity of the RBD-yolk preparation to neutralize WT SARS-CoV-2 in vivo was evaluated. In the Syrian hamster model, animals were treated by nasal administration of 100 µL of the RBD-yolk preparation in a prophylactic (administered 1 h before infection), a post-exposure (administered 1 and 6 h post challenge) or a therapeutic (administered 48, 72 and 96 h post infection) approach (Figure 4A). Irrespective of the administration, infection-induced weight loss was observed in hamsters, with the prophylactic-treated group exhibiting a slightly reduced degree of body weight loss post challenge (Figure 4B). However, no significant differences were identified between the treatments in regard to viral load in nasal washings on 1 to 5 dpc (Figure 4C) or organ loads (Figure 4D) or infectious titers of organ samples (Figure 4E) at the conclusion of the observation period (5 dpc). 

In conclusion, the RBD-antigen preparation induced broadly neutralizing antibodies in chickens that were efficiently transferred into the egg yolk and showed higher titers than the whole-virus preparations. The RBD antigen, in conjunction with multiple boost vaccinations, may serve as a suitable approach to produce reference material for SARS-CoV-2 in vitro diagnostics. However, it was not efficient for in vivo prophylaxis or local diagnostic therapy in the explicit experimental setup.

## 3. Discussion

In this study, we evaluated different SARS-CoV-2 antigens with the objective of inducing specific and functional antibodies in chicken and rabbits. The results indicate that a recombinant protein representing the RBD (319–519) coated on lumazine synthase particles was the most sophisticated antigen. The antigen was found to be highly immunogenic in chickens, with the induced antibodies exhibiting reactivity in multiple immunological assays—including ELISA and IF. Furthermore, these antibodies demonstrated a broad neutralizing activity in vitro against the homologous B1 strain, as well as the BA.1 and BA.5 Omicron VOCs.

Our data indicate that by downsizing the antigen from complete S1 to the RBD, production of neutralizing antibodies was greatly increased. It should be noted that linear epitopes of SARS-CoV-2 spike protein were mapped in the N-terminal domain, subdomain 2, cleavage site, or in the S2 domain, but not in the RBD [17]. However, neutralizing mouse antibodies (nAbs) produced against the SARS-CoV-2 RBD (AA 333–530) describe the receptor-binding motif (RBM) (AA 438–506), located within the RBD domain, as main binding epitopes involving Y449, Y453, L455, F456, F486, Y489, F490, L492, Y495, and Y505 residues. Alternatively, forming an extensive hydrogen-bond network involving Q474, A475, G476, S477, N487, and Y489, stabilized by electrostatic interactions of the light chain with S477, S478, N481 and F486, has been discussed as an important neutralization-sensitive site [18]. While the binding of the RBM directly sterically prevents the binding of the hACE2 receptor, another non-blocking but neutralizing epitope has been described. This is mediated by strong electrostatic interactions involving T345, R346, F347, N440, S443, K444, V445, and N450 [18]. Moreover, an additional conformation-dependent epitope exists that does not impede receptor binding but is capable of neutralizing the receptor (T345 to N450) [18]. Our findings indicate that the RBD (319–519)-coated lumazine synthase particles represent these biologically relevant sites efficiently. 

The selection of chickens for the production of SARS-CoV-2-specific antibodies provides a distinctive approach to harvesting antibodies from the yolk in a non-invasive manner. In addition, the production of IgY from yolk will provide easy access for large quantities of antibodies. The estimated overall production of IgY per year is 30–45 g, based on an average of 300 eggs per year and 100–150 mg of IgY per egg. Approximately 2–10% of these IgY can be antigen-specific [19]. Consequently, IgY antibody preparations can be manufactured in large quantities and subsequently utilized as a reference antibody preparation in commercial test kits, including ELISA, immunofluorescence tests, and antigen-capture assays, such as lateral flow devices [20]. In particular, the batch size of a standard becomes crucial when test results of biological assays like in vitro neutralization are to be compared over an extended period of time and in many different laboratories. The produced RBD antibodies appear to be suitable for such an application, as the antibody preparations demonstrate a broad spectrum of neutralization and are eligible for a variety of different variants. A comparison of sera and yolk preparations indicates that the neutralizing potency of the yolk is lower than that of the corresponding sera. However, when considering the dilution factor of 1:5 of the yolk during the production process it becomes evident that the yolk has at least the same neutralizing capacity as the corresponding sera, e.g., 1:128 VNT100 in yolk vs. 1:512 VNT100 in the corresponding sera for the RBD group. Freeze-dried material can be stored for a prolonged period of time, as shown for SARS-CoV [21], and is available for non-commercial studies/enterprises upon request.

Another potential application of interest is the in vivo application of antibodies for supportive treatment or metaphylaxis of SARS-CoV-2 infection, as suggested by several researchers [3,17,22,23,24,25]. The in vivo treatment with IgY antibodies has been demonstrated to provide protection from infection and to reduce the severity of diseases caused by a number of pathogens, including the Influenza A virus [26,27], Hantavirus [28,29], Ebola virus [30] and other pathogens [31].

In comparison to anti SARS-CoV-2 antibody therapies utilizing monoclonal IgM [3,4], the structural differences between mammalian IgG and avian IgY antibodies may favor the use of IgY in certain applications. For example, IgY is lacking the hinge region, making it less flexible and more stringent [20,32,33]. Furthermore, IgY antibodies do not interact with mammalian or known bacterial FcγR/Fc-binding receptors, thereby reducing the risk of inducing antibody-dependent enhancement (ADE), which can exacerbate infection [33]. Additionally, IgY does not bind to the complement system or rheumatoid factors [33].

However, our preliminary in vivo study with a yolk preparation of the RBD-LS-MPSP against a SARS-CoV-2 challenge infection in a prophylactic (administered 1 h before infection), a post-exposure (administered 1 and 6 h post challenge) or a therapeutic (administered 48, 72 and 96 h post infection) approach did not demonstrate a positive effect with regard to reduction of viral replication or clinical progression (Figure 1). The dosage might have been insufficient, and the delivery method of nasal drops may not have ensured adequate distribution or coverage. Rapid clearance from the nasal mucosa and barriers to antibody absorption could have reduced effectiveness. Additionally, the frequency of administration might have been inadequate to maintain protective antibody levels, and the antibodies may have been unstable in the nasal environment. Furthermore, the viral load introduced during the challenge could have overwhelmed the level of local available antibodies. Further research is needed to explore the full potential of such antibodies.

In conclusion, our findings demonstrate that the SARS-CoV-2 WT RBD domain is immunogenic in chickens and that the generated IgY can be harvested from the yolk over an extended period of approximately 200 days. It has been demonstrated that the produced antibodies exhibit specific reactivity in IFT and are capable of neutralizing SARS-CoV-2 and its VOCs. While there are still some challenges to overcome before considering this method as a potential therapeutic strategy against SARS-CoV-2 infection in animals or humans, our findings indicate that simple intranasal drops are not an effective means of neutralizing SARS-CoV-2 in the sensitive Syrian hamster model.

## 4. Materials and Methods

### 4.1. Preparation of the SARS-CoV-2 Recombinant RBD and S1 Antigen

For expression of the SARS-CoV-2 S1 and the RBD-SD1 domain, amino acids (aa) 17 to 685 or 319 to 519, respectively, were amplified from a codon-optimized synthetic gene (GeneArt synthesis; Thermo Fisher Scientific, Waltham, MA, USA). The constructs were cloned in-frame with an N-terminal modified mouse IgΚ light-chain signal peptide. C-terminally, a 25 aa-linker sequence (GGSQSDSRGGNGNGGGAGGNGGGSA), a SpyTag (AHIVMVDAYKPTK) and a single Strep-Tag (WSHPQFEK) were added. Protein expression and purification was performed as previously described [16]. The SpyCatcher protein was N-terminally fused to the lumazine synthase and expressed in *E. coli*, as previously described [15,34]. The conjugation of the RBD and S1 SpyTag antigens to the LS-MPSP was performed according to a published protocol [15]. Briefly, the antigens and the SpyCatcher-LS were incubated in a molar ratio of 1:2 (antigen:SpyCatcher-LS subunit) in conjugation buffer (20 mM Tris-HCl, 300 mM NaCl, 0.2% Tween at pH 7.5) for 48 h at RT without further processing. Vaccine doses were adjusted with physiological saline to contain 100 µg of conjugated antigen in a total volume of 500 µL per animal.

### 4.2. Preparation of the Inactivated SARS-CoV-2 Whole-Virus Antigen

For SARS-CoV-2 virus-stock generation we inoculated VeroE6 cells with WT (B.1) SARS-CoV-2 isolate (EPI_ISL_406862). Virus containing supernatant of the cell culture was harvested after three days of incubation at 37 °C, having a titer of 10^5.5^ TCID_50_/mL. For inactivation, 1 g of 2-Aminoethyl bromide hydrobromide (Sigma-Aldrich, St. Louis, MO, USA) was dissolved in 50 mL 0.175 M NaOH and incubated for 1 h at 37 °C. From the resulting aziridine solution, 1.5 mL was added to 100 mL of the virus containing cell culture supernatant and incubated for 12 h at 37 °C. The now-inactivated supernatant was transferred into a clean falcon to prevent recontamination. Successful inactivation was evaluated by two cell culture passages on VeroE6 cells, confirming no CPE. This inactivated whole-virus preparation was later used as an antigen for immunization of chickens. In order to increase the concentration of the virus in the inoculum, five 162 cm^2^ cell-culture flasks with confluent grown VeroE6 cells were each inoculated with 100 µL of a SARS-CoV-2 stock. After four days and complete cytopathogenic effect, with no adherent cells remaining, the supernatant was harvested and cell debris was removed by centrifugation at 4550 rcf for 10 min. The resulting 250 mL supernatant was mixed with 28 mL of 1:10 with prediluted (1:200 in cold distilled water) Oxetan-2-one (beta-Propiolactone), leading to an end concentration of 0.05% Oxetan-2-one. Inactivation was completed during 12 h incubation at 4 °C. The treated supernatant was transferred carefully into a clean flask and incubated for 1 h at 37 °C. Successful inactivation was confirmed by two passages on VeroE6 cells without indications of any CPE. The inactivated supernatant was centrifuged at 141,000 rcf for 120 min (Beckman SW 28). The resulting pellets were resuspended in 24 mL DMEM, leading to an approximate 10-fold concentrated pooled-virus preparation.

### 4.3. Immunization of Chickens with Recombinant RBD and S1 Antigen

For antibody and yolk production, female white leghorn chickens (n = 6) were hatched from specific pathogen-free (SPF) chicken eggs (VALO BioMedia, Osterholz-Scharmbeck, Germany) and raised in an S2 quarantine facility until reaching the age of ten weeks. Chickens were kept under free-running conditions with nests and perches. Food and water were provided ad libitum.

For immunization, 100 µg antigens dissolved in 500 µL, were mixed with 100 µL adjuvants (Polygen, MVP adjuvants, Omaha, NE, USA) and administered subcutaneously into the lateral umbilical fold (Plica lateralis). For the recombinant antigens, two subsequent booster immunizations were carried out and blood samples were taken for serological evaluation. Immunizations and sampling were conducted as shown in Figure 1A.

### 4.4. Immunization of Laying Chickens With Inactivated Whole Virus

Female white leghorn chickens (n = 3) were hatched from specific pathogen-free (SPF) chicken eggs (VALO BioMedia, Osterholz-Scharmbeck, Germany) and raised in an S2 quarantine facility until reaching the age of ten weeks. Chickens were kept under free-running conditions with nests and perches. Food and water were provided ad libitum. They were inoculated subcutaneously into the lateral umbilical fold (Plica lateralis) five times with inactivated SARS-CoV-2. Each animal received 2 mL inactivated antigen in DMEM with 10% polygen adjuvant (MVP adjuvants, Omaha, NE, USA). At day 49 and 71. the concentrated virus preparation was used. Immunization and sampling were conducted as shown in Figure 1A.

### 4.5. Immunization of Rabbits with Recombinant RBD and S1 Antigen

Four adult > two-month-old female New Zealand White Rabbits (Crl: KBL (NZW) strain code 052) from a commercial rabbit farm (Charles River Laboratories, Wilmington, MA, USA) were used for immunization. The rabbits were fed with commercial rabbit food (ssniff-Spezialdiäten GmbH, Soest, Germany) and water ad libitum and kept in an S2 quarantine facility. For immunization of rabbits, 100 µg antigen dissolved in 500 µL was mixed with 100 µL adjuvants (Polygen, MVP adjuvants, Omaha, NE, USA) and administered into the musculus quadriceps femoris of the left hind leg. For the recombinant antigens, two subsequent booster immunizations were carried out and blood samples were taken for serological evaluation.

### 4.6. Collection and Preparation of Yolk and Sera

To determine the SARS-CoV-2-specific antibody amount in the yolk, eggs were collected daily from 50 days post prime immunization for the RBD/S1 antigen-immunized animals and from 106 days post prime immunization for the whole-virus immunized animals, as indicated in Figure 1. Therefore, the animals were kept in antigen groups, allowing for the assignment of the eggs to the respective antigen used for immunization, but not on an individual animal level. The yolk was harvested by opening the egg with scissors and tweezers, and removing the allantois fluid by rolling the intact yolk sack on a paper towel. The yolk membrane was then punctured and the yolk was collected into a clean falcon. All the yolk from the same group (RBD, S1 or whole-virus) and from the same week was pooled, resulting finally in 14 pools for the RBD and the S1 group and 13 for the whole-virus group (collection started one week later). The harvested and pooled yolk was prepared for evaluation by mixing it 1:5 (e.g., 1 mL yolk in 4 mL saline) in saline. Mixing was continued until there was a homogenous solution. This solution was frozen (at −20 °C) and thawed three times with proper mixing following every defrosting step. Finally, the homogenized solution was centrifuged, first for 20 min at 2000 rcf and 4 °C and then for 1 min at 13,000 rcf. The liquid phase was separated and stored at −20 °C.

Blood samples were taken by puncturing the V. jugularis or the V. ulnaris, using 0.5 mm syringes, and by collecting the blood into a 2 mL heparin tube (Sarstedt, Nümbrecht, Germany). The blood sample was centrifuged for 10 min at 2000 rcf at 12 °C. The separated serum was collected and incubated at 56 °C for 30 min before storing at −20 °C until use.

### 4.7. RBD-Specific Enzyme-Linked Adsorption Assay (RBD-ELISA)

The ELISA was conducted similarly to as described in [16], as it was shown to be very sensitive and specific: the medium-binding plates (Greiner, Kremsmünster, Austria) were coated with 100 ng/well of the RBD antigen overnight at 4 °C in 0.1 M carbonate buffer (1.59 g Na_2_CO_3_ and 2.93 g NaHCO_3_, ad. 1 L aqua dest., pH 9.6). Thereafter, the plates were blocked for 1 h at 37 °C using 5% skim milk in phosphate-buffered saline (PBS). The sera and yolk samples were diluted 1:100 in Tris-buffered saline with Tween 20 (TBST) and incubated on the coated wells for 1 h at 37 °C. An anti-chicken conjugate (Rabbit anti-Chicken IgY [H + L] Secondary Antibody, HRP; Thermo Fisher Scientific) diluted 1/10,000 was applied. Following the addition of tetramethylbenzidine (TMB) substrate (IDEXX), the ELISA readings were taken at a wavelength of 450 nm on a Tecan Spectra Mini instrument (Tecan Group Ltd., Männedorf, Switzerland), resulting in optical density (OD_450nm_) values. Between each step, the plates were washed three times with TBST. The test was evaluated by 20 negative sera and two dilution rows of known positive samples, and for each run the same defined two replicates of negative (NC) and positive (PC) control sera were carried along.

### 4.8. N-Specific Enzyme-Linked Absorption Assay (N-ELISA)

The samples of the inactivated whole-virus immunized animals were additionally investigated by a commercially available N-protein-based ELISA (ID Screen^®^ SARS-CoV-2 Double-Antigen Multi-species ELISA; Innovative Diagnostics, Grabels, France).

### 4.9. Viruses

For determination of TCID_50_/mL titers, the relevant virus stocks were diluted serially 10-fold and used to inoculate nearly confluent VeroE6 cells (Vero C1008) in two-times 8-fold replicates using two 96-microwell plates. The microwell plates were incubated for 72 h at 37 °C, CPE-positive wells were counted, and virus titer was calculated according to the Sperman–Kaerber formula.

Different SARS-CoV-2 VOCs were used for the live-virus neutralization test: a WT (B.1) isolate (SARS-CoV-2 D614G mutation) (GISAID number: EPI_ISL_406862), an Alpha or B.1.1.7 VOC (EPI_ISL_2131446), a Beta or B.1.351 VOC (GISAID number: EPI_ISL_981782), a Delta or B.1.617.2 VOC (GISAID number: EPI_ISL_1760647), an Omicron BA.1 or B1.1.529 BA.1 VOC (GISAID number: EPI_ISL_6959868) and an Omicron BA.5 VOC (EPI_ISL_12268493).

### 4.10. Virus Neutralization Test (VNT)

The live-virus neutralizing potential of the sera and yolk samples was evaluated in vitro on highly SARS-CoV-2-susceptible VeroE6 cells (Vero C1008) using 96-well plates (Greiner, Kremsmünster, Austria).

At first, sera and yolk samples were diluted two-fold in triplicate and mixed with an equal volume of SARS-CoV-2 containing 100 TCID_50_/mL per well. Following 1 incubation of the sample virus mixture at 37 °C, 100 µL of trypsin-treated cells (trypsinated cells of a confluent 75 cm^2^ cell culture flask in 50 mL DMEM with 2% penicillin/streptomycin) was added to each well. After three days of incubation at 37 °C, the presence of CPE was evaluated using a standard optical transmission microscope. The VNT titer was calculated from three independent dilutions that completely prevented cytopathogenic effect (CPE) in the culture. The virus neutralization titer was then calculated by the formula (−log2) = a/b + c.

The number of cell culture wells without virus replication was (a) divided by the number of cell culture wells per sera dilution (b) plus the −log2 of pre-dilution of the sera/yolk sample (c). The final titer is given as −log2 values of the final sample dilution, able to completely prevent CPE formation on VeroE6 cells infected with 100 TCID_50_/mL.

### 4.11. Indirect Immunofluorescence Assay (IFA)

Confocal microscopy was used to confirm antigen specificity and applicability of generated antibodies. Glass coverslips were located into a 24-well plate (Greiner) and VeroE6 cells were added and infected with 10^2.5^ TCID_50_/well SARS-CoV-2 following 24 h incubation at 37 °C. After another 24 h, incubation cells were fixated with 4% PFA for 30 min at room temperature. Subsequently, cells were permeabilized with 1% triton X100 in PBS for 30 min and blocked with 1% bovine serum albumin in PBS for 30 min. Sera and yolk were diluted 1:200 in PBS, and 400 µL was added per well and this was incubated for 1 h at 37 °C. Following three washing steps with PBST, 400 µL of the secondary antibody 1:400 anti-chicken FITC in PBS was added and incubated for 1 h at room temperature. Following another three washing steps with PBST, glass coverslips were put on a glass side with cell-side down with DAPI containing polyvinyl alcohol, and dried overnight. Images were taken using a confocal fluorescence microscope (Leica TCS SP5, Wetzlar, Germany). Images of Delta and Omicron staining were obtained by non-confocal fluorescence microscope (Nikon Eclipse Ti with pE 300 lite laser, Amstelveen, The Netherlands).

### 4.12. Hamster Experiment

Three groups (prophylactic, post-exposure, and therapeutic) of eight specific pathogen-free Syrian hamsters (*Mesocricetus auratus*, Janvier labs) each were treated intranasally with 100 µL of yolk sample pool of the RBD domain-immunized chickens (the pool of the second week of yolk collecting, which is the second week after the second boost) by pipetting an equally amount of 50 µL directly into the left and right nostril, either 1 h before challenge (prophylactic), or two times (6 and 24 hpc) after challenge (post exposure) or three times (48, 72 and 96 hpc) after challenge (therapeutic). Challenge infection was conducted by applying 35 µL of WT (B.1) isolate (SARS-CoV-2 D614G mutation) (GISAID number: EPI_ISL_406862) into each nostril, with a final titer of 10^4.625^ TCID50/animal. Over a time period of five days, body weight was tracked and nasal washing samples were taken (by flushing 200 µL PBS into each nostril and collecting the reflux into a 2 mL tube) on a daily basis. At day five, animals were euthanized and organ samples were collected for RT-qPCR analysis, together with the nasal washing samples, using the IP4 protocol (https://www.who.int/docs/default-source/coronaviruse/real-time-rt-pcr-assays-for-the-detection-of-sars-cov-2-institut-pasteur-paris.pdf (accessed on 8 July 2024)). By correlating the resulting cq values with a SARS-CoV-2 standard dilution row of known genome copy numbers, the genome copy numbers per mL (gc/mL) were normalized.

## Figures and Tables

**Figure 1 ijms-25-07976-f001:**
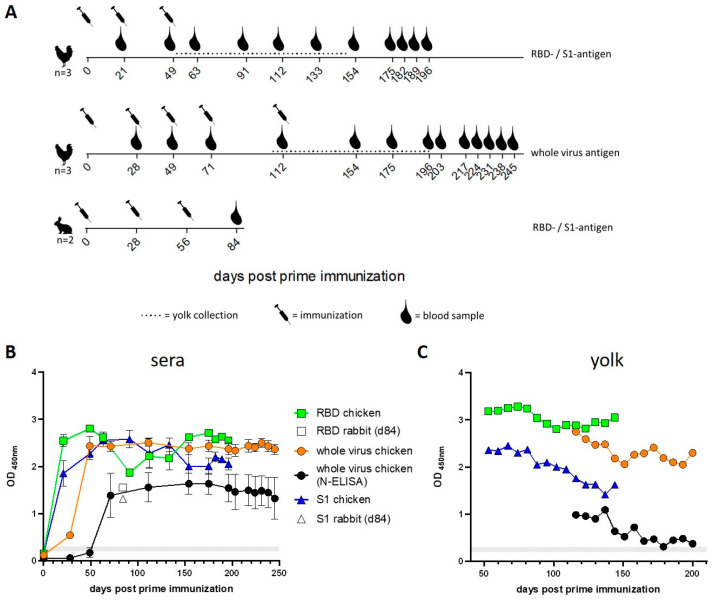
Immune response of chickens and rabbits following immunization with three different SARS-CoV-2 antigen preparations, the RBD antigen (SARS-CoV-2 receptor-binding-domain coupled to LS-particles), the S1 antigen (S1 subunit of the SARS-CoV-2 spike protein coupled to LS-particles) and an inactivated SARS-CoV-2 full-virus preparation. (**A**) The immunization and sampling scheme for the different antigens. (**B**) All sera (dilution factor 1:100) from immunized chickens and reference rabbits were tested by RBD-specific ELISA, and samples from animals immunized with the full-virus preparation were also tested with N-specific ELISA. The mean with SEM of individual animals is depicted for the sera. (**C**) Yolk samples of the chicken were analyzed by RBD-specific ELISA and for the whole-virus group with the N-specific ELISA, in addition. The single values for the weekly yolk-pool samples are shown.

**Figure 2 ijms-25-07976-f002:**
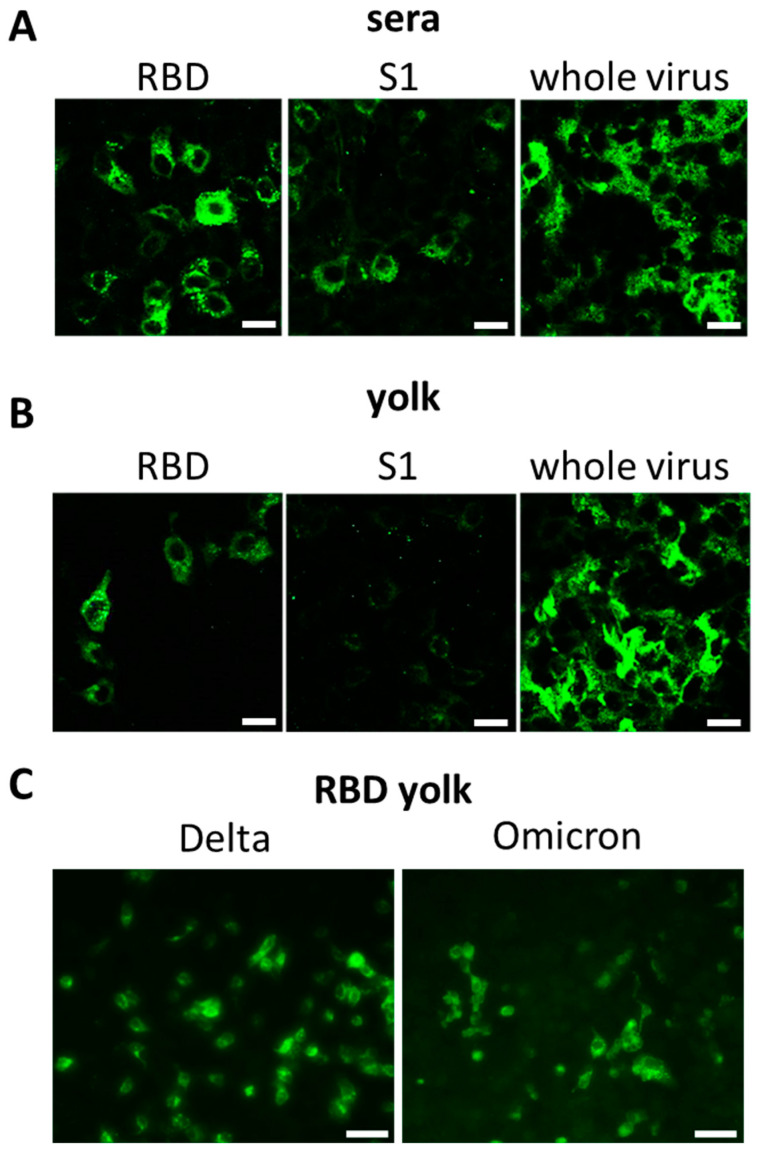
Reactivity of antibody preparations from chickens immunized with either the RBD antigen (SARS-CoV-2 receptor-binding-domain coupled to LS-particles), the S1 antigen (S1 subunit of the SARS-CoV-2 spike protein coupled to LS-particles) or an inactivated SARS-CoV-2 full-virus preparation tested in immunofluorescence assay. (**A**) Serum and (**B**) yolk preparations from chickens are reactive with RBD antigen on immunofluorescence plates showing fluorescence signal of antibody preparations raised against recombinant RBD, S1 or whole-virus preparations. Bar 20 µm. (**C**) Yolk from the RBD group was also positive against the Delta and Omicron BA.1 variant. Bar 100 µm.

**Figure 3 ijms-25-07976-f003:**
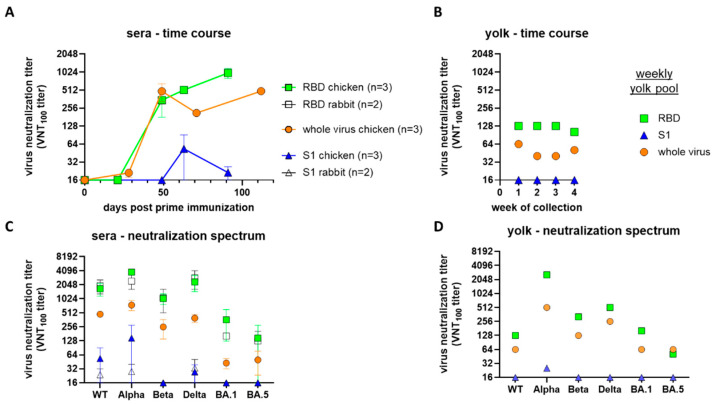
SARS-CoV-2 live-virus neutralization titers (VNT_100_ (titer-neutralizing 100 TCID_50_)) of sera and yolk samples from either the RBD antigen (SARS-CoV-2 receptor-binding-domain coupled to LS-particles), the S1 antigen (S1 subunit of the SARS-CoV-2 spike protein coupled to LS-particles) or an inactivated SARS-CoV-2 full-virus preparation-immunized chickens and rabbits. The course of the neutralizing immune response of chickens in (**A**) sera (mean with SEM) and (**B**) yolk samples (individual values) tested against the sequence-homologous WT variant. Neutralizing reactivity of (**C**) chicken and rabbit sera (mean with SEM) and (**D**) yolk preparations (individual values) were additionally tested against Alpha, Beta, Delta and Omicron BA.1 and BA.5 variants.

**Figure 4 ijms-25-07976-f004:**
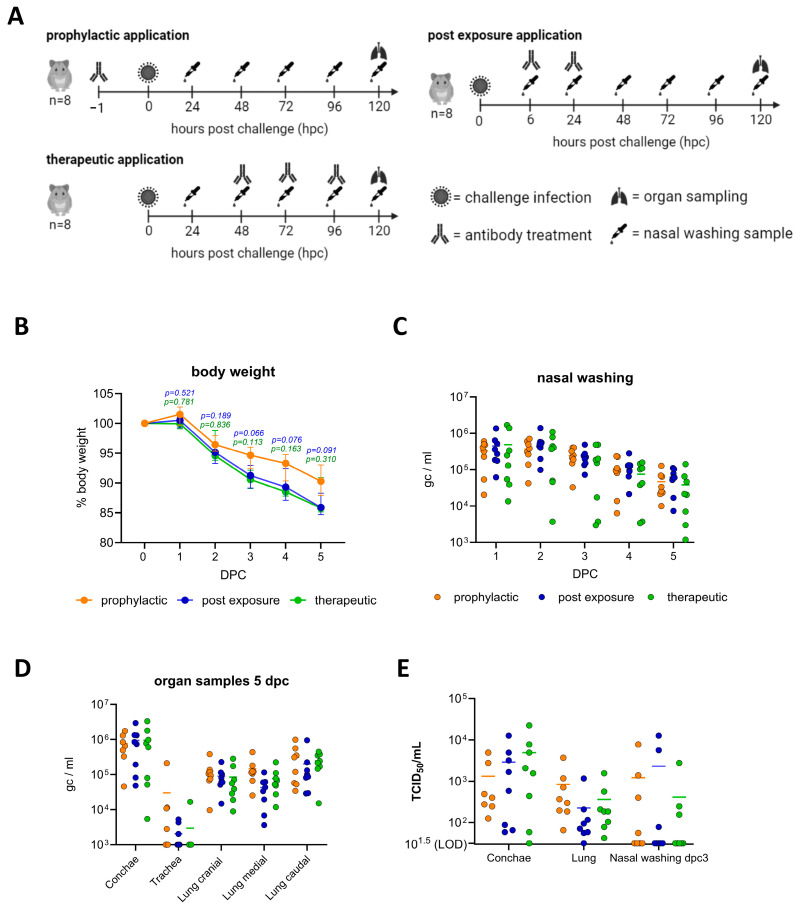
Intranasal application of RBD-yolk preparation in a prophylactic, therapeutic or post-exposure experimental setup in Syrian hamsters. (**A**) Experimental setup. (**B**) Relative body weight. (**C**) Virus genome (genome copies per mL) in nasal washing samples and (**D**) 5 dpc organ samples. (**E**) Infectious virus quantified in conchae, lung (cranial) and nasal washing samples of 3 dpc. Statistical differences were calculated by two-way ANOVA with Tukey´s multiple comparison test.

**Table 1 ijms-25-07976-t001:** SARS-CoV-2 live-virus neutralization titers of sera and yolk samples from RBD-, S1- or whole-virus-immunized chickens and rabbits.

log2 VNT100 titer						
			**virus used in neutralization assay**
**antigen**	**species**	**biological replicate**	**WT**	**Alpha**	**Beta**	**Delta**	**BA.1**	**BA.5**
**RBD**	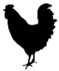	#1	11.3	12	10.3	12.3	9.7	8.7
#2	10.7	11.7	9	10.3	4	4
#3	9.7	12	10.3	10.7	8	4
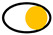	yolk pool	7	11.3	8.3	9	7.3	5.7
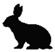	#1	11.7	11.3	11.7	10.7	12	7.3
#2	10.7	10.3	10.7	9	10.7	7.3
**S1**	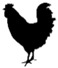	#1	4	4	4	4	4	4
#2	4	4	4	4	4	4
#3	7	8.7	4	5.7	4	4
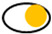	yolk pool	4	4.7	4	4	4	4
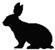	#1	4	4	4	4	4	4
#2	5.3	5	5.3	4	5.7	4
**whole virus**	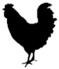	#1	9	6.3	8.3	6.3	7	5
#2	8.7	5	9.7	5	8	5
#3	9	8.7	8.7	8.7	9	7.7
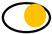	yolk pool	6	9	7	8	6	6

Log_2_ VNT titer of ≤4 = negative/below tested threshold	
Log_2_ VNT titer of >4 and ≤7 = positive at a moderate level	
Log_2_ VNT titer of >7 and ≤10 = positive at a high level	
Log_2_ VNT titer of >10 = positive at a very high level	

## Data Availability

Data is contained within the article.

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
