# Peer review of "Evaluation of SARS-CoV-2-Specific IgY Antibodies: Production, Reactivity, and Neutralizing Capability against Virus Variants"

_ijms, 2024, doi:10.3390/ijms25147976_

Round 1

Reviewer 1 Report

Comments and Suggestions for Authors

The manuscupte submitted to Int J Mol Sci by Christian Grund and co-authors is devoted to the analysis of IgY from RBD-immunised chickens.

Some issues should be resolved by the authors

1) A series of experiments were carried out on chickens, rabbits and hamsters. The text does not contain any information about the animals, their living conditions, age, weight or other ethical aspects.

2) How do the authors presume the non-invasive collection of antibodies from eggs (lines 15-16)? Please note that the animal from which the blood is taken does not die comparing to when something is taken from the egg, a chick will not be born...

3) Why are there two graphs in Figure 1-C that start after day 100?

4) Was the generation of antibodies against lumazine synthase been tested using this accessory antigen?

5) Some of the figures related to testing the protective properties of IgY (lines 180-213) should be moved to the main text of the article, as the authors refer to these results in the Discussion and Abstract. In this case, it is incorrect to refer to them as Supplementary.

In general, the article gives the impression that the material has been collected into a whole without a main idea, which greatly reduces the value of the article. I suggest that the authors formulate of the hypothesis that they have tested in their experiments.

The title is also unfortunate, since the characteristic "broad neutralisation" is completely irrelevant to the results obtained and their possible physiological significance.

Comments on the Quality of English Language

To improve the text of the article, I recommend hiring a qualified English editor.

Author Response

Comment1: A series of experiments were carried out on chickens, rabbits and hamsters. The text does not contain any information about the animals, their living conditions, age, weight or other ethical aspects.

Response1:

We have added additional information about the animals and their living conditions:

4.3. Immunization of chickens with recombinant RBD- and S1-antigen

For antibody and yolk production female white leghorn chickens (n=6) were hatched from specific pathogen free (SPF) chicken eggs (VALO BioMedia, Germany) and raised in a S2 quarantine facility until reaching the age of ten weeks. Chickens were kept under free-running conditions with nests and perches. Food and water were provided ad libitum.

For immunization 100 µg antigens dissolved in 500 µL, were mixed with 100 µL adjuvants (Polygen, MVP adjuvants, USA) and administered subcutaneously into the lateral umbilical fold (Plica lateralis). For the recombinant antigens two subsequent booster immunizations were done and blood samples were taken for serological evaluation. Immunizations and sampling were conducted as shown in Figure 1A.

4.4. Immunization of laying chickens with inactivated whole virus

Female white leghorn chickens (n=3) were hatched from specific pathogen free (SPF) chicken eggs (VALO BioMedia, Germany) and raised in a S2 quarantine facility until reaching the age of ten weeks. Chickens were kept under free-running conditions with nests and perches. Food and water were provided ad libitum. They were inoculated subcutaneously into the lateral umbilical fold (Plica lateralis) five times with inactivated SARS-CoV-2. Each animal received 2 mL inactivated antigen in DMEM with 10 % polygen adjuvant (MVP adjuvants, Omaha, USA). At day 49 and 71 the concentrated virus preparation was used. Immunization and sampling were conducted as shown in Figure 1A.

4.5. Immunization of rabbits with recombinant RBD- and S1-antigen

Four two-month-old female New Zealand White Rabbit-rabbits (Crl: KBL (NZW) strain code 052) from a commercial rabbit farm (Charles River Laboratories) were used for immunization. The rabbits were fed with commercial rabbit food (ssniff-Spezialdiäten GmbH, Germany) and water ad libitum and kept in a S2 quarantine facility. For immunization of rabbits 100 µg antigen dissolved in 500 µl, were mixed with 100 µl adjuvants (Polygen, MVP adjuvants, USA) and administered into the musculus

quadriceps femoris of the left hind leg. For the recombinant antigens two subsequent booster immunizations were done and blood samples were taken for serological evaluation.

Ethical approvals:

Institutional Review Board Statement: Immunizations were approved by the responsible ethics committee of the State Office of Agriculture, Food Safety, and Fishery in Mecklenburg– Western Pomerania (LALLF M-V) and gained governmental approval under the registration numbers LVL M-V/TSD/7221.3-2-042/17. Hamster experiments were evaluated by the responsible ethics committee of the State Office of Agriculture, Food Safety, and Fishery in Mecklenburg– Western Pomerania (LALLF M-V) and gained governmental approval under the registration numbers LVL MV TSD/7221.3-1-067/21.

Comment 2: How do the authors presume the non-invasive collection of antibodies from eggs (lines 15-16)? Please note that the animal from which the blood is taken does not die comparing to when something is taken from the egg, a chick will not be born...

Response 2: We would like to argue that the collection of antibodies from eggs is non-invasive because it avoids the physical intrusion and stress associated with blood sampling from rabbits or cattle. This method leverages the natural process of egg-laying, which does not harm the hen, making it a more humane and ethical approach to antibody harvesting.

Comment 3: Why are there two graphs in Figure 1-C that start after day 100?

Response 3: The reason the two graphs of the whole virus immunized chickens group started after day 100 post-prime immunization, rather than at day 49 post-prime immunization as with the other antigens, is that yolk sampling was initiated at a later time point. The main intention with the inactivated whole virus group was to establish a strong positive control; therefore, the chickens were immunized four times before starting to collect the yolk samples.

Comment 4: Was the generation of antibodies against lumazine synthase been tested using this accessory antigen?

Response 4: The generation of antibodies with specificity against lumazine synthase was not evaluated. However, given the good reactivity observed against the coated antigens, we did not feel it necessary to test for an immune reaction against the lumazine synthase particle. Nevertheless, such testing would be theoretically possible by LS-ELISA and was conducted by Okba et al. 2020, so that it can be assumed that also antibodies against the LS-particle are generated.

Okba, N. M. A., Widjaja, I., van Dieren, B., Aebischer, A., van Amerongen, G., de Waal, L., … Haagmans, B. L. (2020). Particulate multivalent presentation of the receptor binding domain induces protective immune responses against MERS-CoV. Emerging Microbes & Infections, 9(1), 1080–1091. https://doi.org/10.1080/22221751.2020.1760735

Comment 5: Some of the figures related to testing the protective properties of IgY (lines 180-213) should be moved to the main text of the article, as the authors refer to these results in the Discussion and Abstract. In this case, it is incorrect to refer to them as Supplementary.

Response 5: We agreed on the suggestion to move the supplementary figure to the main text.

Comment 6: In general, the article gives the impression that the material has been collected into a whole without a main idea, which greatly reduces the value of the article. I suggest that the authors formulate of the hypothesis that they have tested in their experiments.

Response 6: We rephrased the abstract and the introduction to clarify the aim of the study, which was to 1) check the ability of the antigens to induce a humoral immune response and 2) to characterize the immune response and 3) validate if and for long it is possible to harvest these highly specific antibodies from the yolk.

Comment 7: The title is also unfortunate, since the characteristic "broad neutralisation" is completely irrelevant to the results obtained and their possible physiological significance.

Response 7: We adapted the title as follows: Evaluation of SARS-CoV-2 Specific IgY Antibodies: Production, Reactivity, and Neutralizing Capability Against Virus Variants

Reviewer 2 Report

Comments and Suggestions for Authors

The manuscript presents the possibility of long-term production of IgY antibodies specific for SARS-CoV-2 after immunization of laying hens, using a preparation of the whole inactivated virus and the recombinant RBD domain and the S1 subunit domain presented on the LS-MPSP protein particle using the SpyCatcher/SpyTag technology. A comparison of the immune response between rabbits and birds to immunization with the RBD and S1 LS-MPSP constructs is also presented. An intranasal vaccination experiment was also carried out using hamsters.

Comments:

The study was conducted on a small number of laying hens n=3 and rabbits n=2. Therefore, it would be appropriate to indicate in the title that this is a preliminary or pilot study. This number should also be provided in the abstract.

Figure S1 and Table S1 may be moved to the manuscript because they are helpful for discussion.

Line 47-50 – only the benefits are highlighted, but not recognizing the Fc fragment also has negative consequences in the mechanism of the protective response. Please take this into account.

Line 56-73 – This part is appropriate for discussion, not introduction. In turn, there is no clearly defined purpose of the study that was intended to be achieved.

Line 97-98 – “In contrast, the booster immunization on day 28 with whole virus antigen resulted in a pronounced increase in the antibody response.” Antibody response (Reactivity) or Antibody level ? Throughout the article, please clearly distinguish the level of antibodies from their reactivity (affinity, avidity, neutralization).

Line 345 – no number of chickens used.

Line 361 – no number of rabbits used.

Why are hamsters not included in Fig 1A?

Comments on the Quality of English Language

Minor editing of English language required

Author Response

Comment 1: The study was conducted on a small number of laying hens n=3 and rabbits n=2. Therefore, it would be appropriate to indicate in the title that this is a preliminary or pilot study. This number should also be provided in the abstract.

Response 1: We have added the exact animal number per group to the abstract.

Comment 2: Figure S1 and Table S1 may be moved to the manuscript because they are helpful for discussion.

Response 2: We agreed on the reviewer comment and therefore moved the supplementary figure and table to the main text.

Comment 3: Line 47-50 – only the benefits are highlighted, but not recognizing the Fc fragment also has negative consequences in the mechanism of the protective response. Please take this into account.

Response 3:

To also highlight potential disadvantages of a missing Fc segment we modified the statement accordingly:

Unlike mammalian IgG, IgY does not activate the complement system, avoiding inflammatory responses and assay interference [12]. Nevertheless while IgY antibodies offer several advantages such as reduced risk of antibody-dependent enhancement (ADE) {Fink, 2017 #49} and lack of interaction with mammalian Fc receptors [12], these can also limit their effectiveness by reducing immune system engagement [12], potentially shorter half-life, and impairing their ability to clear pathogens effectively [12].Line 56-73 – This part is appropriate for discussion, not introduction. In turn, there is no clearly defined purpose of the study that was intended to be achieved.

We restructured and rephrased the whole introduction to better describe the scientific question we evaluated.

Comment 4: Line 97-98 – “In contrast, the booster immunization on day 28 with whole virus antigen resulted in a pronounced increase in the antibody response.” Antibody response (Reactivity) or Antibody level ? Throughout the article, please clearly distinguish the level of antibodies from their reactivity (affinity, avidity, neutralization).

Response 4: We rephrase the antibody response to antibody level and checked the wording through the whole manuscript.

Comment 5: Line 345 – no number of chickens used.

Response 5: In addition to the Figure 1A the animal number is now mentioned in the methods and material part too.

Comment 6: Line 361 – no number of rabbits used.

Response 6: In addition to the Figure 1A the animal number is now mentioned in the methods and material part too.

Comment 7: Why are hamsters not included in Fig 1A?

Response 7: In Figure 1A we only visualized the experimental setup for the immunization leading to the antibodies, which were later used for the passive immunization of the hamsters. Nevertheless, the experimental setup of the hamster passive immunization trial is visualized in Figure 4A now. We hope the reviewer find it useful.

Reviewer 3 Report

Comments and Suggestions for Authors

Authors collected IgY containing egg yolk from the immunized chickens were highly specific in ELISA and IFA and had broad virus neutralizing capacity against the wild type (WT) SARS-CoV-2 (B.1) and several variants of concern (VOC) including the Alpha 68 (B.1.1.7), Beta (B.1.351), Delta (B.1.617.2), Omicron (B.1.1.529) BA.1 and BA.5. However, yolk preparations derived from the RBD LS-MPSP immunized chickens were insufficient in initial in vivo protection studies using the Syrian hamster model. Further galenic approaches are needed to further explore the potential of egg yolk preparations for prophylactic, post-exposure or as supportive therapeutic approach.

Although this manuscript is interesting, several issues arise.

1.     It is negative results in hamster model, although the RBD antigen preparation induced broadly neutralizing antibodies in chicken that were efficiently transferred into the egg yolk and showed higher titers than the whole virus preparations.

2.     It is helpful to discuss the reason for non-protecting this animal from SARS-CoV-2.

3.     It is useful to measure the behavior of these antibodies in hamster model.

4.     Why did authors select the RBD antigen of SARS-CoV-2.

5.     Figure 1. There are many abbreviations. It is helpful to explain those abbreviations in legend.

6.     Please explain RBD, S1 and RBD yolk in Figure 2 legend. No explain for C.

7.     What is unit of Y axis in Figure 3?

Author Response

Comment 1: It is negative results in hamster model, although the RBD antigen preparation induced broadly neutralizing antibodies in chicken that were efficiently transferred into the egg yolk and showed higher titers than the whole virus preparations.

Comment 2: It is helpful to discuss the reason for non-protecting this animal from SARS-CoV-2.

Response to 1 and 2: 

We further discussed the reason for non-protecting of the IgY in the hamster model:

However, our preliminary in vivo study with a yolk preparation of the RBD-LS-MPSP against a SARS-CoV-2 challenge infection in a prophylactic (administered 1 hour before infection), a post-exposure (administered 1 and 6 hours post challenge) or a therapeutic (administered 48, 72 and 96 hours post infection) approach did not demonstrate a positive effect with regard to reduction of viral replication or clinical progression (Figure 1). The dosage might have been insufficient, and the delivery method of nasal drops may not have ensured adequate distribution or coverage. Rapid clearance from the nasal mucosa and barriers to antibody absorption could have reduced effectiveness. Additionally, the frequency of administration might have been inadequate to maintain protective antibody levels, and the antibodies may have been unstable in the nasal environment. Furthermore, the viral load introduced during the challenge could have overwhelmed the l level of local available antibodies. Further research is needed to explore the full potential of such antibodies.

.Comment 3: It is useful to measure the behavior of these antibodies in hamster model.

Response 3: We agree on reviewers comment that it would be from great interest to measure antibody kinetics in hamster animal model.

Comment 4: Why did authors select the RBD antigen of SARS-CoV-2.

Response 4: The receptor binding domain of the SARS-CoV-2 spike protein is mediating the binding of the natural main receptor hACE2. Therefore, specific antibodies are potentially neutralizing.

Comment 5: Figure 1. There are many abbreviations. It is helpful to explain those abbreviations in legend.

Response 5: We adapted the figure legend accordingly.

Comment 6: Please explain RBD, S1 and RBD yolk in Figure 2 legend. No explain for C.

Response 6: We adapted the figure legend accordingly.

Comment 7: What is unit of Y axis in Figure 3?

Response 7: We now specified in the Y-axis of the figure and in the figure legend that we conducted a virus neutralization test using a 100 TCID50, as described in the method section in detail. In general, it’s a common way to show neutralizing titers, which is given as ration without a unit. Example are: https://www.nature.com/articles/s41467-020-20247-4, 10.1016/j.heliyon.2024.e31026, https://doi.org/10.1016/j.xcrm.2020.100040

Round 2

Reviewer 1 Report

Comments and Suggestions for Authors

The manuscript has been significantly revised

Reviewer 3 Report

Comments and Suggestions for Authors

Revisecd manuscript has been sufficiently improved. I have no further comment.